# Elucidation of the low-expressing erythroid CR1 phenotype by bioinformatic mining of the GATA1-driven blood-group regulome

Ping Chun Wu [1], Yan Quan Lee [1], Mattias Möller [1,2], Jill R. Storry [1,3] & Martin L. Olsson [1,3] ✉

Genetic determinants underlying most human blood groups are now clarified but variation in expression levels remains largely unexplored. By developing a bioinformatics pipeline analyzing GATA1/Chromatin immunoprecipitation followed by sequencing (ChIP-seq) datasets, we identify 193 potential regulatory sites in 33 blood-group genes. As proof-of-concept, we aimed to delineate the low-expressing complement receptor 1 (CR1) Helgeson phenotype on erythrocytes, which is correlated with several diseases and protects against severe malaria. We demonstrate that two candidate *CR1* enhancer motifs in intron 4 bind GATA1 and drive transcription. Both are functionally abolished by naturally-occurring SNVs. Erythrocyte *CR1*-mRNA and CR1 levels correlate dose-dependently with genotype of one SNV (rs11117991) in two healthy donor cohorts. Haplotype analysis of rs11117991 with previously proposed markers for Helgeson shows high linkage disequilibrium in Europeans but explains the poor prediction reported for Africans. These data resolve the longstanding debate on the genetic basis of inherited low CR1 and form a systematic starting point to investigate the blood group regulome.

Blood group antigens are polymorphisms of proteins, glycoproteins, and glycans on red blood cells (RBCs), which can stimulate antibody formation and evoke subsequent transfusion reactions in patients transfused with incompatible blood. The genetic bases for blood group polymorphisms are largely resolved, but the molecular mechanisms regulating antigen expression are far less clear. The first discovered and most well-known example is the Fy(a−b−) (Duffy-negative) phenotype identified in individuals of West African origin, caused by a single nucleotide variant (SNV) in the promoter region of *ACKR1* that disrupts a GATA1-binding site and abolishes the expression of ACKR1 on RBCs[1]. A polymorphism in the GATA1 recognition sequence in *ABO* intron 1 was later discovered to limit GATA1-binding, resulting in weak A or B antigen[2]. More recently, an SNV in a GATA1-binding site that lies between the *XG* and *CD99* loci has been shown to regulate the expression of both genes, abolishing the expression of *XG*

and reducing the expression of *CD99*[3,4]. A deeper understanding of blood group antigen regulation enables reliable genotyping to ensure transfusion safety since persons with weakly-expressed antigens may be erroneously phenotyped as antigen-negative. Among the blood group systems, some are more prone to variation in expression, making phenotyping unreliable. A well-known example is the Knops antigens on complement receptor 1 (CR1; CD35) which carry such a risk as levels of CR1 on RBCs vary greatly among individuals[5].

CR1 is a single-pass, type I membrane glycoprotein found on erythrocytes, leukocytes, dendritic cells, and glomerular podocytes[6–8]. Human CR1 exhibits three types of polymorphism: 1) expression levels, i.e. copy number of CR1 molecules per RBC[5], 2) peptide sequence variants resulting in different antigens of the Knops system[9], and 3) four structural polymorphisms with molecular weight ranging from 160 kDa to 250 kDa due to variation in number of long homologous

[1]Division of Hematology and Transfusion Medicine, Department of Laboratory Medicine, Lund University, Lund, Sweden. [2]Department of Clinical Genetics and Pathology, Office for Medical Services, Region Skåne, Lund, Sweden. [3]Department of Clinical Immunology and Transfusion Medicine, Office for Medical Services, Region Skåne, Lund, Sweden. ✉e-mail: Martin_L.Olsson@med.lu.se

repeats[10]. CR1 is an important complement regulatory protein and functions by binding to and removing C4b- and C3b-bearing immune complexes[11,12]. Low erythroid CR1 levels have been correlated with gallbladder carcinomas, systemic lupus erythematosus (SLE), sarcoidosis, and Alzheimer's disease[13–15]. CR1 also serves as a ligand for the *Plasmodium falciparum* erythrocyte membrane protein 1 (PfEMP1) expressed on infected RBCs[16]. The interaction of PfEMP1 with CR1 on uninfected RBCs results in rosette formation and exacerbates infection and disease severity[17,18]. RBCs with low CR1 copy number rosette only poorly, and thus this phenotype confers protection against severe malaria[19].

The number of CR1 molecules expressed on RBCs exhibits a 10-fold variation across individuals; for most people, it lies within a range of 100–1,000 molecules per cell[5,13]. However, if the RBCs carry fewer molecules (20–100), the individuals are considered to be of the Helgeson phenotype[20]. Since persons of the Helgeson phenotype have barely detectable amounts of erythrocyte CR1 by hemagglutination, they are considered to be of the serological null phenotype[21].

The Helgeson phenotype was reported with a prevalence of 1%, both in people of African American and Caucasian origins[22]. A *Hin*dIII restriction site polymorphism (rs11118133:A > T) in intron 27 of *CR1* was first found to be a predictor for low erythrocyte CR1 expression in Caucasians, but this site was later shown to exhibit poor predictive power in African Americans[5,23,24]. Moreover, this *Hin*dIII restriction site did not correlate to the CR1 expression levels on neutrophils or B cells[5]. Later, genome-wide association studies and the 1000 Genomes project discovered other SNVs (which were strongly associated to the *Hin*dIII restriction site) to also be predictors for low CR1 expression, such as rs2274567:A > G (p.His1208Arg) and rs3811381:C > G (p.Pro1827Arg)[14,25–27]. However, no functional explanation was provided as to why CR1 expression varies, nor were they better predictors than the *Hin*dIII site for the Helgeson phenotype outside of the Caucasian population.

Given the strong influence of GATA1 for the expression of ABO, FY, and XG blood groups, we set out to develop a systematic bioinformatics-based pipeline, using chromatin immunoprecipitation sequencing (ChIP-seq) data to examine regulatory factors controlling human blood group antigen expression (here designated the blood group regulome). We, therefore, started by collating and screening multiple publicly available GATA1 ChIP-seq datasets in order to apply them to the genes implicated in the 36 blood group systems that had been ratified by the International Society of Blood Transfusion (ISBT) when this project was initiated[28]. Given the well-established variability of CR1 expression[29], we decided to use this as our primary proof-of-principle example for the pipeline and set out to understand the molecular and genetic basis of the Helgeson phenotype. We hypothesized that the inherited variability observed is regulated at the transcriptional level and is cell-lineage specific. In this work, the *CR1* locus was interrogated for polymorphisms in potential regulatory regions that could explain the great difference in interindividual expression levels of CR1. We thereby resolve the longstanding debate on the genetic basis of inherited low CR1 and provide a systematic starting point to explore the blood group regulome.

## Results

To elucidate the GATA1-driven blood group regulome, we processed individual ChIP-seq experiments in human primary erythroblasts through a custom-made in silico analysis pipeline using a combination of public bioinformatics tools and in-house scripts (Fig. 1). We identified a total of 193 GATA1-binding sites of which 114 exhibited overlapping peaks in all four datasets selected based on strict quality criteria (see Methods). The 193 candidate targets localized to 33 genes of 36 blood group systems and are summarized in Supplementary Data 1, with a range of 1 to 15 candidate target motifs per gene. Whilst the list of targets resulting from our GATA1-focused pipeline constitutes an open-access starting point for a deeper understanding of the blood group regulome, here we aimed to solve the long-standing enigma about the functional basis of the Helgeson phenotype. As proof of principle for our bioinformatics pipeline approach, we turned our attention to the six potential GATA1-binding sites that were assigned to *CR1*. The aim was to propose a reliable genetic marker predicting CR1 expression levels on RBCs, based on experimental validation of the identified candidate motifs.

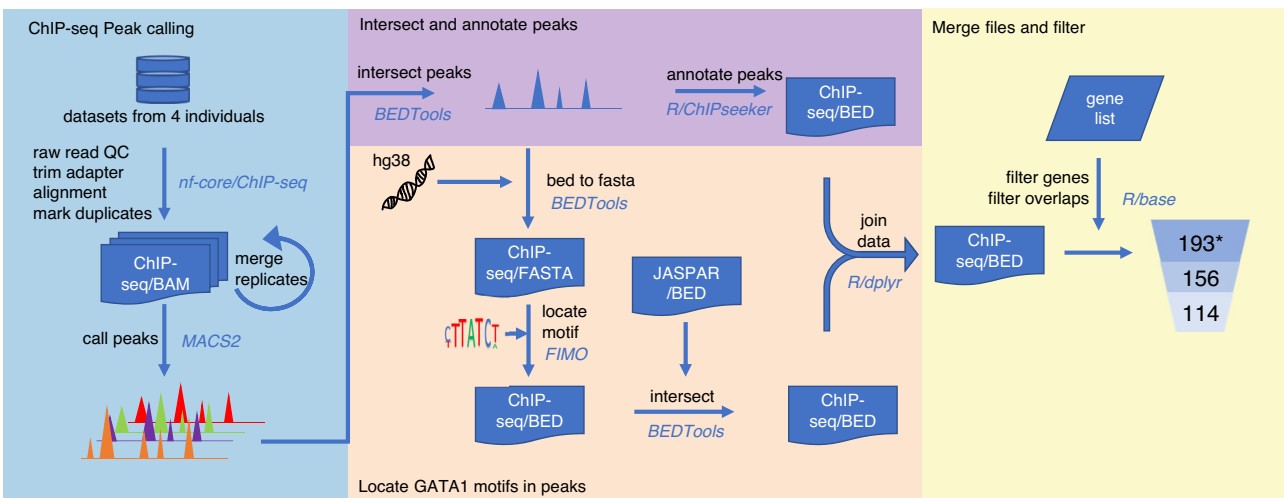

**Fig. 1 | Analysis pipeline of four GATA1 ChIP-seq datasets on primary erythroblasts.** Raw sequencing data were extracted from publicly available databases (n = 4) and subjected to the above analysis pipeline adapted from nf-core for ChIP-seq analysis. Functions for each analysis step are shown next to the corresponding arrows along with the software/packages used (italicized, blue text). The format of the files processed in each step is noted in the flowchart (white text on blue background). Background colors represent the four major steps in the data processing. After all analysis and data filtering steps, 193 GATA1 binding sites were found across 33 blood group-related genes, including 6 for *CR1*. *The pipeline predicted 193 sites with peaks that overlapped in at least two datasets including the reference dataset (defined as the dataset containing the most peaks), 156 sites with overlapping peaks in at least three datasets including the reference, and finally 114 sites that were found in all four datasets.

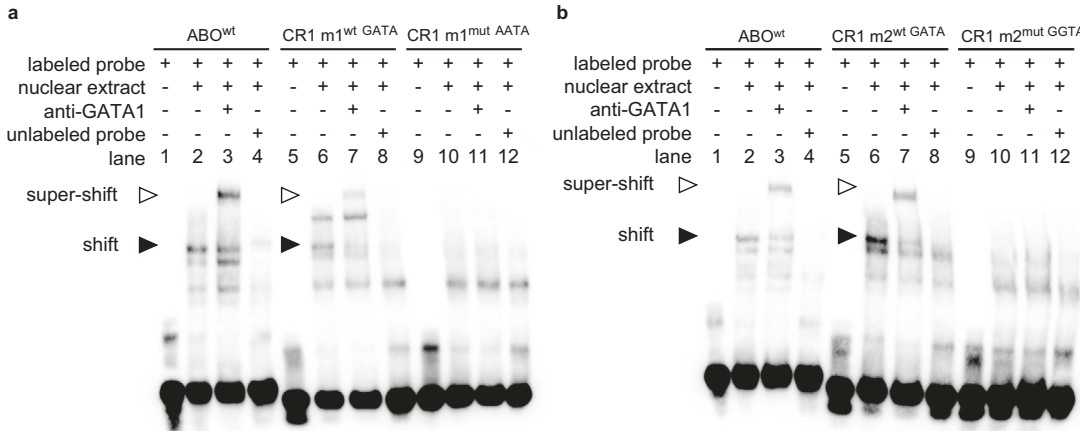

**Fig. 2 | GATA1 binds to two predicted motifs in *CR1* intron 4 by Electrophoretic Mobility Shift Assays (EMSA).** EMSAs were performed with 39-bp biotinylated probes corresponding to the consensus sequences of the predicted GATA1 binding motifs and their naturally-ocurring variants. **a** Motif 1 with the corresponding mutant rs368344378:G > A (motif altered from GATA to AATA). **b** Motif 2 with the corresponding mutant rs11117991:T > C (motif altered from GATA to GGTA). The GATA1 binding site located in *ABO* intron 1 was used as a positive control for both assays. Only the wildtype motifs exhibited a shift (black arrow ▶, lanes 2 and 6) upon incubation with nuclear extract from K562 cells, and a super-shift (white arrow ▷, lanes 3 and 7) with the addition of anti-GATA1, highlighting its affinity for GATA1. Preincubation with 200-fold unlabeled probes (lanes 4 and 8) abolished the mobility shifts, indicating specificity. The results shown are representative of three independent experiments. m1: motif 1; m2, motif 2; wt, wild type; mut, mutant.

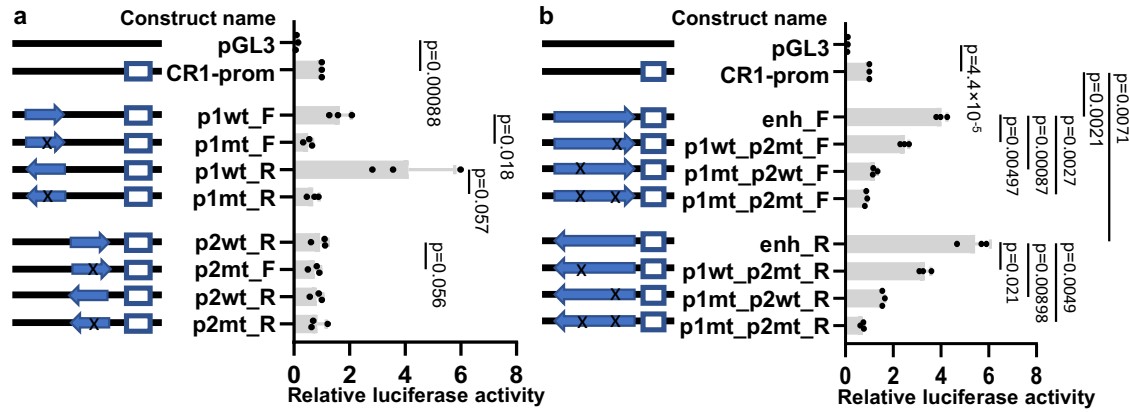

**Fig. 3 | Motif 1 or a combination of motifs 1 and 2 enhanced expression in a luciferase assay.** HEL cells were transfected with pGL3 plasmids carrying wild-type or mutant motifs 1 and 2 upstream of a luciferase gene driven by the *CR1* promoter. Firefly luciferase activity was normalized to a co-transfected *Renilla* luciferase control and displayed relative to pGL3 carrying only the *CR1* promoter. Data represent mean values; error bars indicate standard errors of the mean. The relevant segments of the plasmid constructs are shown on the left of each panel where x indicates the mutant sequence (not to scale). **a** *CR1* motifs 1 and 2 assessed in separate constructs. **b** *CR1* motifs 1 and 2 are assessed as conjoined motifs. CR1-prom, promoter of *CR1* (displayed as an open blue box to the left of the diagram); enh, enhancer (displayed as a blue arrow); p1, peak1; m2, peak 2; wt, wild type; mt, mutant. F and R indicate forward and reverse directions respectively relative to the *CR1* gene direction. All samples were run in technical triplicates and were repeated in three independent experiments. The ratio between firefly and *Renilla* luciferase luminescence values were calculated, and the means were compared using paired *T* test (two-sided). Source data are provided as a Source Data file.

## Functional investigation of the top candidate GATA1-binding sites in CR1

Based on FIMO and JASPAR motif scores for GATA1, peak scores from ChIP-seq peak calling, and the numbers of overlapping peaks from different datasets at these loci, two of the six sites were selected for further functional characterization. The two sites (hereafter named motifs 1 and 2) are located in *CR1* intron 4, 710 bp apart. Each site had one reported SNV each that directly disrupts the predicted GATA1-binding motif: rs368344378:G > A in motif 1, and rs11117991:T > C in motif 2, with total minor allele frequencies (MAF) as reported in gnomAD v3.1.2 of $1.31 \times 10^{-5}$ and 0.157, respectively[30]. In silico analysis of the binding energy scores from JASPAR for the wild type and variant sequences at both motifs indicated the loss of functional GATA1-binding capacity (motif 1 wild type = 0.887 (relative score), motif 1 variant rs368344378:G > A = 0.763; motif 2 wild type = 0.933, motif 2 variant rs11117991:T > C = 0.808). Experimentally, both motifs demonstrated binding of GATA1, and subsequent loss of binding when mutated to the minor allele in electrophoretic mobility shift assays (EMSAs) (Fig. 2). A luciferase reporter assay showed that when the two motifs were examined separately, both elevated transcription significantly. However, only motif 1 displayed a significant decrease in activity when mutated to the minor allele (Fig. 3a). Interestingly, when both motifs were present in the same construct, a significant increase in luciferase activity was detected regardless of the orientation of the motifs relative to the reporter gene (Fig. 3b), consistent with enhancer status. Decreased activity was observed when either motif was disrupted, and the most significant decrease occurred when mutating both motifs. Moreover, the wild-type motifs together contributed to a

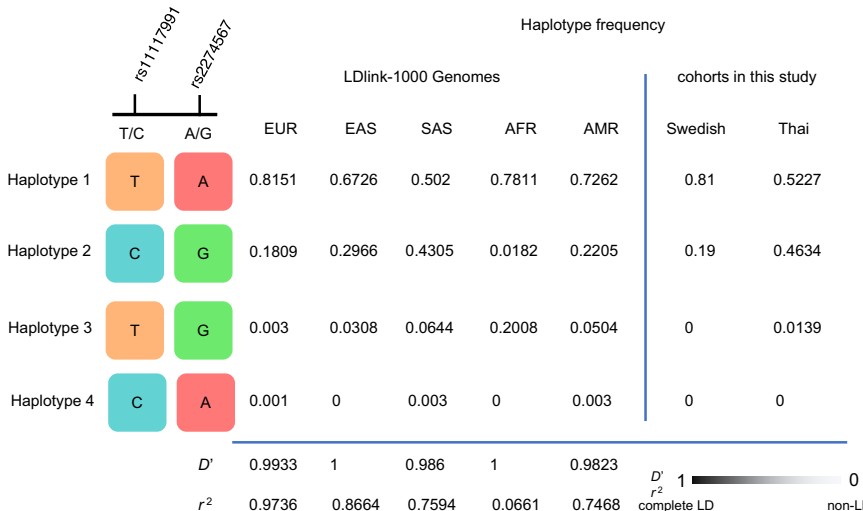

**Fig. 4 | Haplotype frequencies and LD parameters of rs11117991 and rs2274567 observed from LDlink and the Swedish and Thai cohorts in this study.** Haplotype frequencies were obtained from LDlink by LDhap, or calculated for the Swedish and Thai cohorts reported in this study (Swedish, $n = 100$; Thai, $n = 396$). Linkage disequilibrium (LD) indexes $D'$ and $r^2$ were calculated or obtained from LDlink by LDpair, where 1 indicates complete linkage and 0 indicates no linkage between the two loci. If any of the haplotypes is 0, the index $D'$ will have a value of 1. EUR, European; EAS, East Asian; SAS, South Asian; AFR, African; AMR, Ad Mixed American. The four colors represent the four different nucleotides at the two loci discussed.

greater enhancement of luciferase activity that exceeded the addition of the results when the two motifs were tested separately, which suggests these two closely located motifs work as synergistic enhancers.

## Cohorts and haplotypes

Based on the reported approximate frequencies, we hypothesized rs11117991 (MAF: 0.157) to be a more likely reason underlying the Helgeson phenotype. We, therefore, genotyped two cohorts of different geographic/ethnic backgrounds and compared rs11117991:C with two previously proposed Helgeson markers (rs2274567:G and the earliest marker, the *Hin*dIII restriction polymorphism, rs11118133:T). These markers were previously reported as a predictor for low expression of CR1 on RBCs[27]. Genotyping of Swedish blood donor samples showed that rs11117991 (motif 2) exhibited complete linkage disequilibrium (LD) with rs2274567 (Supplementary Table 1). However, in a Thai cohort, we observed incomplete linkage of the two loci, allowing separation of rs11117991:C from rs2274567:G in 11/396 samples (Supplementary Table 2), potentially explaining the poor prediction of Helgeson using previous markers in non-Caucasians. The polymerase chain reaction-restriction fragment length polymorphism (PCR-RFLP) analysis with *Hin*dIII performed on the Thai cohort showed complete concordance with rs2274567, consistent with a previous study[27].

The haplotype frequencies in the cohorts shown in Fig. 4 were similar to those reported on LDlink (data from the 1000 Genomes project), with the Swedish cohort comparable to the European superpopulation and the Thai cohort intermediate between the South and East Asian superpopulations[31]. The measures of LD, $D'$ and $r^2$, revealed that the European superpopulation exhibited the strongest linkage between rs11117991 (motif 2) and rs2274567 with $D' = 0.9933$ and $r^2 = 0.9736$. The other superpopulations showed weaker LD between these loci, with the weakest found in the African superpopulation, $D' = 1$ and $r^2 = 0.0661$. Haplotype frequencies between rs11117991 and SNVs determining the Knops blood group antigens were also examined with LDlink but none of these SNVs were exclusively linked to the minor allele of rs11117991 (Supplementary Fig. 1).

Archived gDNA from the index case of the Helgeson phenotype[21] was available for analysis, and as predicted by our model, the sample genotyped as homozygous for the minor alleles at rs11117991, rs2274567, and rs11118133.

## CR1 expression correlates to the rs11117991 genotype

Given the genotypic correlation, we then investigated the expression of CR1 by mRNA studies and different protein detection methods. CR1 on the RBC surface showed a dosage effect that correlated with both the rs11117991 and rs2274567 genotypes when assessed by flow cytometry in the Swedish blood donors. Samples from donors homozygous for the major allele expressed significantly more CR1 than donors heterozygous or homozygous for the minor allele (p-values = 0.000023 and 0.00092, respectively) (Fig. 5a). Since rs2274567 was in complete LD with rs11117991 in the Swedish samples, the contributions of the two loci could not be separated. Consequently, a subset of samples from the Thai cohort ($n = 41$) was selected to resolve which of the two is the functionally important marker (Supplementary Table 2 and Fig. 4). The minor allele of both loci correlated significantly with low expression of CR1 (Fig. 5b, c). Graphically, the data suggested a dosage effect of the rs11117991 genotype (Fig. 5b) but not with rs2274567 (Fig. 5c), although this could not be verified statistically. Therefore, we performed multiple regression analysis which revealed that the mean fluorescence intensity (MFI) was significantly correlated with the rs11117991 genotype (p-value = 0.0441) but not rs2274567 (p-value = 0.2683) (Fig. 5d).

We noted variation of expression within the same genotype, most pronounced in the rs11117991:T/T genotype (Fig. 5a, b). To investigate if other SNVs in this region within intron 4 of *CR1* contributes to the variation observed, the same 952-bp fragment as used in the luciferase constructs was amplified and sequenced in seven of the lowest CR1-expressing samples with rs11117991:T/T genotype. However, three samples did not show any deviation from the *CR1* reference sequence (NG_007481) and no other plausible explanations underlying the low expression in the rs11117991:T/T group was revealed in the four remaining samples (Supplementary Note in the Supplementary Information).

Immunoblotting of RBC membrane proteins with anti-CR1 on selected representative samples from the Swedish and Thai cohorts (Fig. 6, original immunoblots available in the Source Data file) supported a functional role for rs11117991 genotype, consistent with the flow cytometry data.

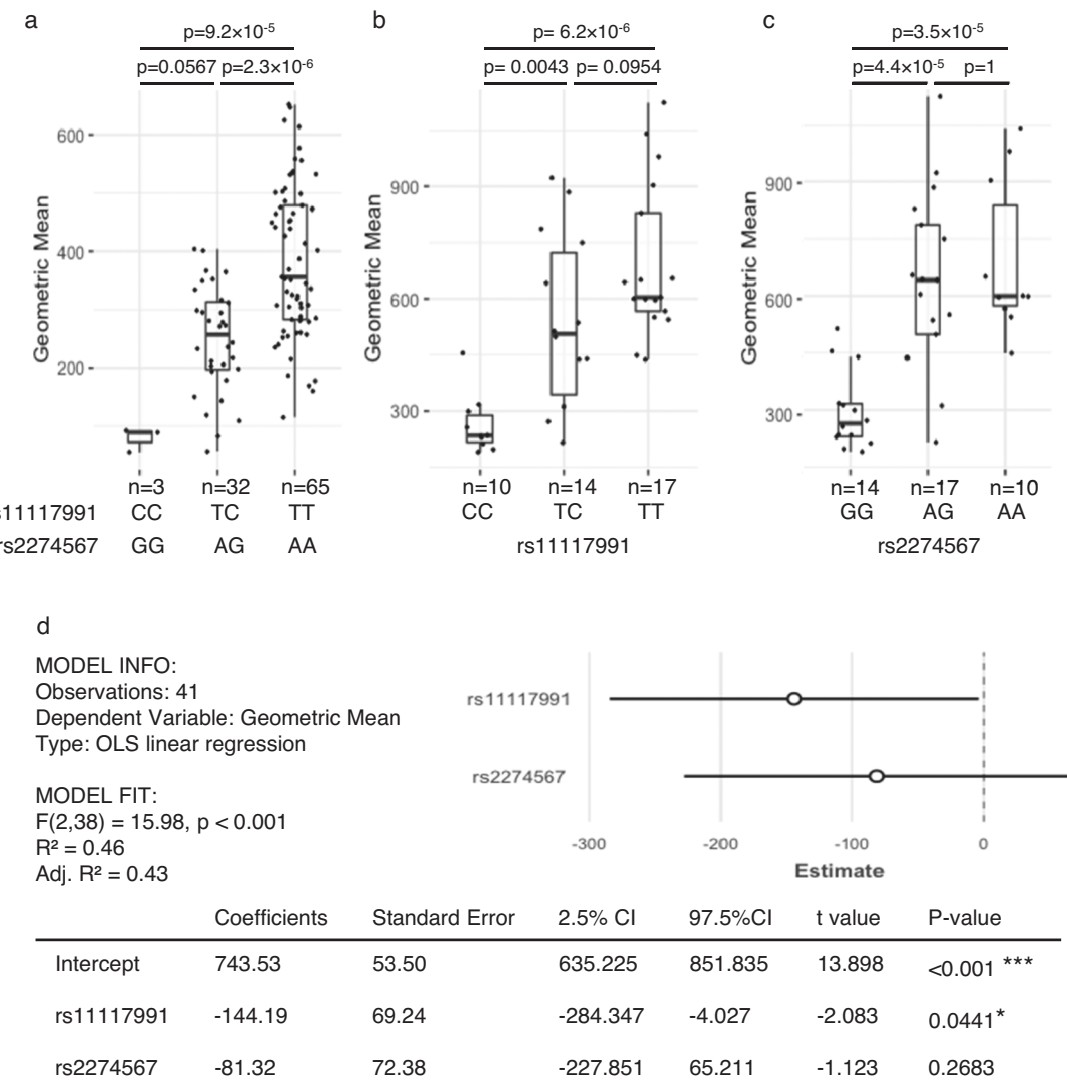

**Fig. 5 | CR1 expression on the RBC surface is correlated to the rs11117991 genotype in both sample cohorts.** Flow cytometry with PE-conjugated anti-CD35 with **a** the Swedish cohort ($n = 100$) performed on fresh RBCs grouping by both rs11117991 and rs2274567 genotype, and **b** a subset of the Thai cohort ($n = 41$) on thawed RBCs when grouped by rs11117991 genotype, and **c** grouped by rs2274567 genotype. Geometric mean fluorescence intensities (MFI) were compared when grouped by either the rs11117991 or rs2274567 genotypes. For the Swedish cohort, rs11117991 and rs2274567 genotypes were in perfect concordance and could not be separated. Boxplots indicate median and 1st and 3rd quartiles and error bars indicate values within 1.5 times of the interquartile range; dots outside the bars indicate values greater than 1.5 times the interquartile range. For **a**–**c**, the MFI was compared between each genotype group with one-way ANOVA and post-hoc test with the Bonferroni method. **d** Multiple linear regression analysis of the Thai cohort with geometric MFI as the dependent variable and the genotypes of rs11117991 and rs2274567, as independent variables with ordinary least squares (OLS) linear regression. The coefficients and the 95% confidence intervals (CIs) are plotted against the dependent variable. Variables were assigned according to genotype as follows: the homozygous major was designated as 0, heterozygous as 1, and homozygous minor as 2. No additional adjustments were made for multiple comparisons. *P*-values: *, <0.05; ***, <0.001. Source data are provided as a Source Data file.

Measurement of *CR1* transcript levels in the same Thai donor subset revealed a similar picture with a trend towards a dosage effect, wherein transcript levels increased with the number of rs11117991:T alleles. This was not observed with rs2274567 (Fig. 7a). Multiple regression analysis showed *CR1* transcript levels to be significantly correlated with rs11117991 (*p*-value = 0.00937), but not with rs2274567 (*p*-value = 0.95316) (Fig. 7b).

## Discussion

Transcriptional regulation has been shown to play an important role in the RBC surface expression of several different blood group antigens[1,2,32–35], however, the understanding of how our blood group genes are regulated is incomplete. In fact, the molecular basis of the interindividual variation in RBC expression levels observed for most blood groups is unknown, although genotype/phenotype discrepancies in the ABO and FY systems resulted in the discovery of SNVs affecting GATA1-binding sites crucial for RBC expression of these blood groups[1,2]. Later, the presence or absence of Xg[a] expression was shown to correlate with a SNV altering a GATA1 site located between, and co-regulating, two homologous genes, *XG* and *CD99*[3,4]. Similarly, polymorphisms have been demonstrated in binding sites of other transcription factors, e.g. in the binding motifs for RUNX1 and EGR1 in *A4GALT* that govern the expression of the P1 blood group antigen[34,35], and in a TAL1 binding site in *SMIM1* that governs expression of the Vel blood group[33]. Furthermore, mutations in transcription factors themselves can also cause changes in blood group phenotypes as shown for KLF1[36,37]. Once identified, such regulatory polymorphism can be incorporated in modern approaches for genomic typing[38,39]. Here, we report a systematic approach to predict and validate regulatory sites in blood-group-encoding genes through the bioinformatic mining of

 

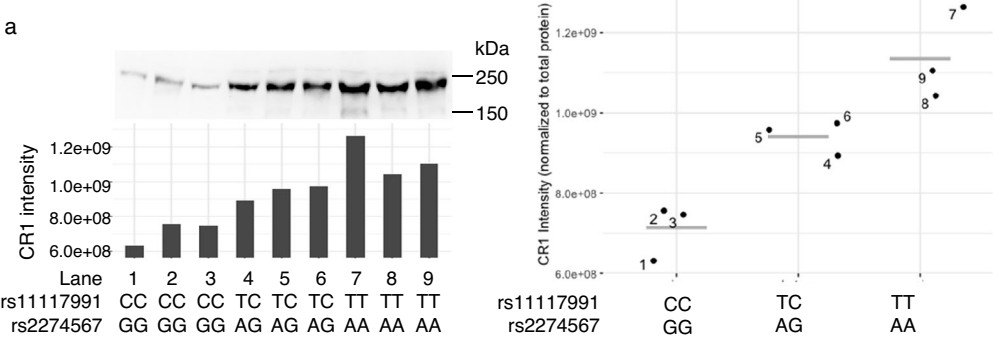

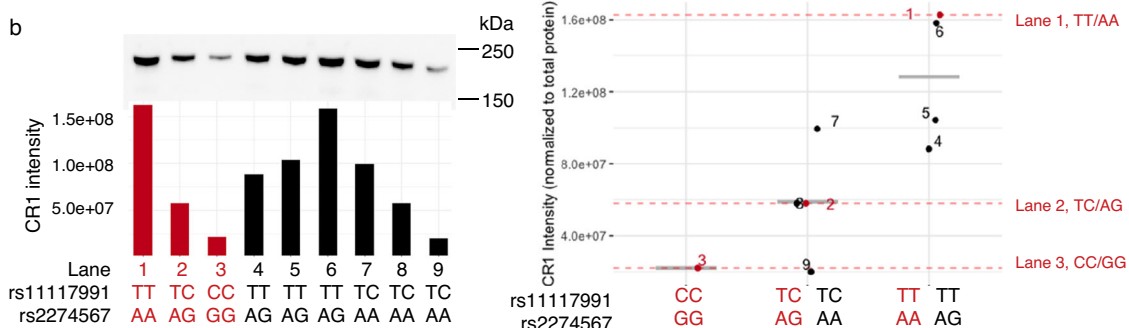

**Fig. 6 | CR1 protein levels correlate to genotypes grouped by rs11117991 in both cohorts.** CR1 protein level was measured by western blot with anti-CD35 (clone E11) in RBC ghost membranes, the intensity of the band for CR1 is normalized to the total protein from the respective individual lane and showed under the blot as bar charts. Dot plots on the right showed the individual measurement from the left, each dot is labelled to the corresponding lane, and the grey bar indicates the arithmetic mean of the respective genotype group. **a** Western blot of CR1 in the Swedish cohort. CR1 expression is shown to correlate according to the genotypes both for rs11117991 and rs2274567 where the two SNVs are inseparable. **b** Western

blot of CR1 in Thai cohort. The intensities for the control samples (lanes 1–3, shown in red) are also indicated on the dot plot on the right as dashed lines with their respective rs11117991 and rs2274567 genotypes. The mean of the samples from lanes 4, 5, and 6 is closer to lane 1, whereas the mean of the samples from lanes 7, 8, and 9 falls nearly on top of lane 2. kDa, kilodaltons. Both blots were cropped for clarity. For both cohorts, one successful replicate was performed independently. The complete blots with positive and negative controls as well as all source data are provided in a Source Data file.

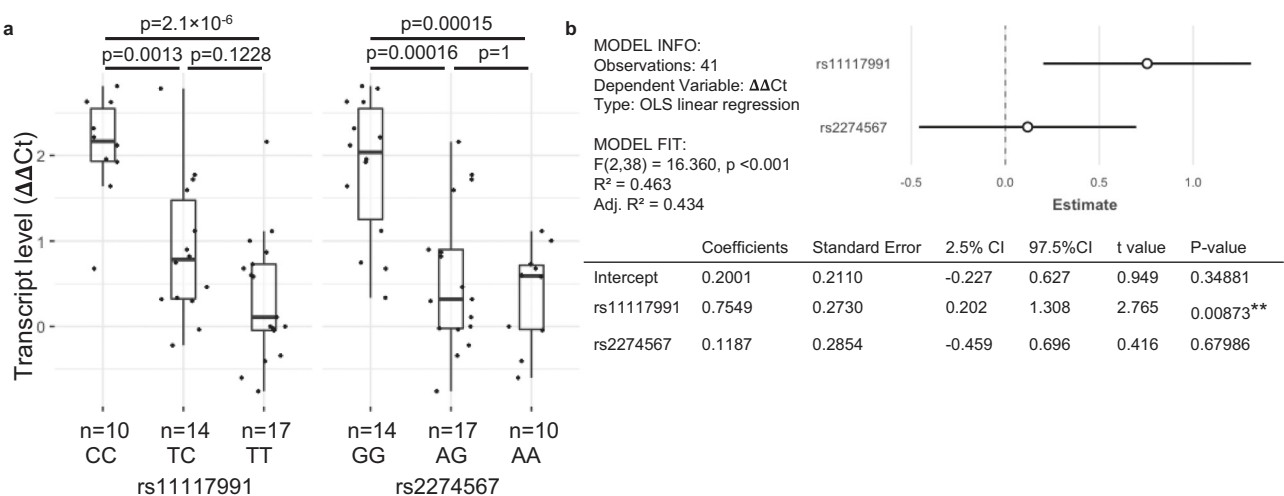

**Fig. 7 | _CR1_ transcript levels are correlated to the rs11117991 genotype in the Thai cohort.** Investigation of _CR1_ transcript levels was performed on thawed, leukodepleted RBCs in a subset of the Thai cohort (n = 41). **a** ΔΔCt were compared (normalized to _GAPDH_ and an interplate control). Boxplots indicates median and 1st and 3rd quartiles and error bars indicate values within 1.5 times of the inter-quartile range; dots beyond the bars denote values greater than 1.5 times the interquartile range. Technical triplicates were performed for each sample and were successful. **b** Multiple regression analysis with ΔΔCt as the dependent variable and

genotypes of rs11117991 and rs2274567 as independent variables with ordinary least squares (OLS) linear regression. The coefficients and the 95% confidence intervals (CIs are plotted against the estimated dependent variable ΔΔCt. Variables were assigned according to genotype as follows: the homozygous major was designated as 0, heterozygous as 1, and homozygous minor as 2. No additional adjustments were made for multiple comparisons. _P_-values: **, <0.01. Source data are provided as a Source Data file.

existing GATA1 ChIP-seq datasets from public databases. In Supplementary Data 1, all the resulting 193 candidate GATA1-binding motifs are listed for further study. As proof-of-concept, we focused our attention on one of the blood groups that varies the most in its interindividual RBC expression levels, which are also correlated to clinically important diagnoses like malaria, SLE and Alzheimer's disease. We thereby uncovered a regulatory locus in the *CR1* gene and demonstrated, through in vitro experiments, cohort studies, and further data mining, that a SNV at this locus causes the very weakly CR1-expressing Helgeson phenotype, exerting its effect by disrupting a GATA1-binding motif. This study, therefore, provides evidence that resolves a longstanding enigma that has persisted in the field for over half a century[21].

Based on our data, we not only provide a genetic marker for determining Helgeson status but also show the underlying mechanism to involve transcriptional downregulation. We originally identified six potential GATA1-binding sites and noticed, after careful inspection of various bioinformatically defined parameters, that two motifs in close proximity in intron 4 of *CR1* stood out based on motif scores, peak scores, and the numbers of overlapping peaks among the experiments in the datasets interrogated. We confirmed that both the wildtype motifs 1 and 2 indeed bind GATA1 and lose binding when the minor variants rs368344378:A and rs11117991:C, respectively, are introduced (Fig. 2). Interestingly, it became apparent that motif 1 is essential to this *CR1* enhancer's activity, since a single-base disruption of its GATA site entirely abrogated any enhancement, as demonstrated in the luciferase assay (Fig. 3a). In conjoined constructs with both motifs 1 and 2, motif 2 exhibited a modulatory effect; disruption of the GATA site in motif 2 suppressed but did not eliminate enhancer activity (Fig. 3b). Notably, activity was reduced to promoter background levels when both motifs were mutated, indicating that no additional factors in the inserted 952-bp region are able to boost transcription, at least not in the HEL/luciferase system. Since the minor allele frequency at 0.157 of the SNV in motif 2 (rs11117991:C) was a better fit with the prevalence of the Helgeson phenotype at about 1 %, we suspected that rs11117991 could be the long-sought genetic determinant that could enable genetic prediction of erythroid CR1 levels. The other four GATA1-binding sites were also checked for their biological variant allele frequencies in gnomAD, but all were exceedingly low (1−3 alleles out of a total of >150,000 alleles in all superpopulations), and did not fit the reported Helgeson prevalence. Therefore, they were not investigated further. While CR1 is known to have an important role in malaria the variants affecting these four candidate motifs do not appear to have undergone evolutionary selection in malaria-affected populations. Our functional characterization of the prioritized GATA motifs were performed in well known in vitro models but may not correspond fully to the in vivo situation. Therefore, we continued our study by investigating the relation of these and previously proposed genetic markers to CR1 expression on RBCs from healthy blood donors.

After having confirmed that the original Helgeson sample was indeed homozygous for rs11117991:C, we expanded our investigation to two cohorts where RBCs, DNA, and mRNA were available for study and from two populations with a clear difference in the allele frequencies of interest. In both the Swedish and Thai cohorts, we identified a significant correlation between the minor allele of rs11117991 and low CR1 expression on RBCs by flow cytometry and immunoblotting. Since there was complete LD between rs11117991 and a previously reported predictor, rs2274657, in the Swedish cohort, we were unable to dissect the contribution of each locus. However, LD was incomplete in the Thai cohort, which permitted the identification of 11 samples that carried haplotypes not observed in the Swedish samples. This allowed us to separate the effects of polymorphisms at each locus. By multiple regression analysis (Figs. 5d and 7b), we demonstrated that CR1 expression at both protein (Fig. 5) and transcript (Fig. 7) levels correlated well to rs11117991 genotype but not to

rs2274567. We considered the possible need for additional cohorts beyond Caucasians, e.g. because the previously proposed *Hind*III predictor was first reported to fail in individuals of African origin. However, due to the low frequency of rs11117991:C in Africans and the limited availability of fresh samples from healthy donors, we concluded that the Thai cohort sufficed to prove the functional importance of rs11117991. Instead, we resolved this issue through linkage analysis to provide an explanation why the prediction in Africans had been poor. Through the haplotype frequencies of rs11117991 and rs2274567 and using $D'$ and $r^2$ as parameters for LD (Fig. 4), it became clear that rs11117991 is in the highest LD with rs2274567 in the European superpopulation ($D' = 0.9933$ and $r^2 = 0.9736$). This explains why the genetic prediction of low RBC expression of CR1 was most accurate in Caucasians using rs2274567 or other substitute markers exhibiting high LD with rs11117991. In sharp contrast, the linkage between these two loci is lost in the African superpopulation ($r^2 = 0.0661$), which therefore diminishes the predictive accuracy using surrogate markers. This also corroborated a previous observation that the *Hind*III restriction site does not correlate with CR1 expression levels on RBCs in Africans[24]. Also, the haplotype frequencies and LD parameters from both our cohorts correlate well with the corresponding geographic superpopulations from the 1000 Genomes project as calculated by LDlink.

Although we report here that rs11117991 is an important determinant for CR1 expression on RBCs and consequently also a potential disease marker for those diagnoses in which low erythroid CR1 has been reported, it is apparent that CR1 expression varies greatly even within each genotype group, as exemplified in Fig. 5. Thus, we cannot exclude that there could be other (positive or negative) regulatory elements contributing to the *CR1* gene expression levels outside the loci studied here. In fact, regulation of the *CR1* gene was examined earlier and it was reported that neither the 5' nor 3' untranslated region is correlated to the expression levels[40]. However, this did not eliminate other intronic regions of *CR1* or long range cis-regulatory element that may regulate gene expression and could be controlled by other transcription factors besides GATA1 are thereby not included in this study. Therefore, more work may be needed to understand further how *CR1* is regulated beyond GATA1. Moreover, CR1 has been suggested to be one of the top targets that is heavily influenced by malaria pressure. This has given rise to many variants in the coding sequence of CR1 and thereby altered antigen expression, not least among individuals of African descent. It is therefore possible that alternative or complex regulatory mechanisms of CR1 remain to be elucidated in other populations.

The minor allele frequencies of rs11117991:C in both East Asian and South Asian superpopulations of the 1000 Genomes project are higher than that of the Europeans. It has been suggested that the low-expression of CR1 is protective against severe malaria[41,42], thus higher frequency of the minor allele in Asia could be the result of selection pressure in malaria-endemic regions. However, in the African superpopulation there was not a higher minor allele frequency despite Sub-Saharan Africa being an endemic region[43]. This could be explained by a *CR1* polymorphism resulting in the loss of one of the Knops antigen, Sl[a], which has a prevalence of approximately 30% in African Americans and 68–72% in West Africans but is rare in Caucasians[17,19]. Sl(a−) RBCs exhibit poorer rosette formation than Sl(a+) cells[17], although soluble CR1 fragments containing blood group variants including Sl(a+) vs. Sl(a−) showed no functional differences in assays for inhibition of RBC rosette formation or invasion by *P. falciparum*[44]. Moreover, other Knops antigen phenotypes such as KCAM− or McC(b+) have been associated with reduced risk of severe malaria[45,46]. It appears therefore, that two principally distinct mechanisms to modify CR1 have evolved to avoid severe effects of malaria; quantitatively by reducing CR1 expression on RBCs and qualitatively by altering the antigen portfolio in the Knops blood group system. Interestingly, studies of CR1

expression indicated that the quantitative approach was more commonly observed in Asian populations like Taiwanese, Cambodians or Papua New Guineans when compared to Africans from Mali[19,47]. However, these studies were largely based on the *Hin*dIII restriction site (rs11118133) or other genetic markers that do not form the actual mechanistic basis for the Helgeson phenotype. Our finding that rs11117991 determines CR1 expression allows for genotyping of the Helgeson phenotype and more precise investigation of its relation to malaria.

It has been proposed that the Helgeson phenotype might be due not only to low density of CR1 on RBCs but also to the absence of high-prevalence Knops system antigens[48]. Following the discovery of rs11117991 as the functional determinant, homozygosity for which results in the Helgeson phenotype, we challenged this idea by performing a linkage analysis of haplotypes involving the seven SNVs implicated in all Knops antigens known to date (Supplementary Fig. 1). Our conclusion is that rs11117991:C is not exclusively linked to any of the SNVs encoding high-prevalence KN antigens. Since the Helgeson phenotype is characterized by very low CR1 on RBCs, it is likely that limitations in sensitivity of detection for these high-prevalence antigens may well have given rise to false negative serological results. In analogy, the same phenomenon also delayed the discovery of the most recently published addition to the Knops system. The SNV determining the new DACY and YCAD antigens (rs2274567:A/G, p.His1208Arg)[49] was actually described decades ago as a marker for low CR1 expression[27]. Thus, rs2274567 defines these antithetical antigens and its minor allele is tightly linked with rs11117991:C, which makes YCAD hard to detect serologically[49]. Conversely, our linkage analysis explains the findings in a previous study in which RBCs from individuals carrying the low-prevalence Sl2/Vil or McC^b antigens were associated with higher CR1 levels[50]. In both cases, the minor alleles encoding these antigens are only found in *cis* with the major, high-expressing allele of rs11117991 (Supplementary Fig. 1). Even if the Helgeson genotype is not linked to alloantibody formation against CR1 in general, its tight linkage to certain Knops-antigen-defining exonic SNVs makes low CR1 expression a feature often observed in the corresponding antigen-negative individuals.

In summary, we present here a robust bioinformatics pipeline that allowed us to generate predictions of GATA1 regulatory sites in blood group genes, using publicly available data. As a first proof of principle, we translated the in silico findings for *CR1* into in vitro validation experiments and cohort studies, leaving many other predicted targets to be explored. Our results demonstrate that an abolished GATA1-binding site at rs11117991 constitutes the mechanism underlying the very low CR1 expressing Helgeson phenotype on RBCs, reported over 50 years ago. When active, this locus enhances CR1 expression together with another nearby GATA1-binding site at rs368344378. We also describe how haplotype frequencies of rs11117991 are linked with previously reported Helgeson predictors and how this varies in different populations, explaining why these markers including the *Hin*dIII restriction site, were unreliable as predictors of the Helgeson phenotype except in Europeans. With the genetic determinant established, a DNA-based test for the Helgeson phenotype is now made possible. This has the potential to improve the diagnostic toolbox for malaria, SLE and Alzheimer's disease, in which susceptibility or severity is correlated to low levels of CR1 on RBCs.

## Methods

This study complies with all relevant ethical regulations, including the ethics approval obtained from the University of Phayao Human Ethics Committee in Thailand.

### Blood and DNA samples

Anonymized, EDTA-anticoagulated blood samples collected as part of the routine blood donation procedure were obtained from 100 random healthy blood donors at the Department of Clinical Immunology and Transfusion Medicine, Office for Medical Services, Region Skåne, Sweden. According to the Swedish research law, using waste/excess fully anonymized and/or pooled biological material does not require ethics approval. Thus, no approval was required for this study, apart from the permission quoted below for the Thai cohort, and from the Dept. of Clinical Immunology and Transfusion Medicine for the Swedish cohort to obtain the anonymized blood from waste material (ref. no. 2018:22 and 2020:16). For the Thai cohort of healthy blood donors, acid-citrate-dextrose-anticoagulated samples were obtained from the Lampang Hospital and Saraburi Hospital, Thailand, following ethics approval (UP-HEC 2/024/59) by the University of Phayao Human Ethics Committee, Thailand and informed consent. DNA from the index case of the Helgeson phenotype was kindly provided by Life-Share Blood Center (Shreveport, TX, USA) from their archived rare sample inventory and cannot be redistributed. However, it is also available via the Serum Cells and Rare Fluid exchange program (SCARF). Materials from the donor cohorts cannot be redistributed in accordance with the respective permits.

### ChIP-seq analysis and annotation

We obtained ChIP-seq read data in FASTQ format through the public databases Cistrome and ENCODE for GATA1, a transcription factor known to be important for in blood group expression and erythropoiesis[51,52]. Inclusion criteria for dataset selection were: ChIP-seq experiments performed on adult human primary (pro)erythroblasts; detection of peaks for known GATA1-binding sites in *ABO* and *ACKR1*; and, if retrieved from Cistrome, valid quality controls as defined in the database. The GATA1 ChIP-seq experiments included in this study are summarized in Supplementary Data 2. Codes are available in the GitHub repository (https://github.com/ILM-MLOlab/ChIP-seq-GATA1, https://zenodo.org/badge/latestdoi/501283332). The ChIP-seq datasets were first subjected to Nextflow(nf)-core pipeline analysis (https://github.com/nf-core/chipseq, version 1.0.0), which incorporates multiple software to perform quality control, adapter trimming, and alignment, and to identity duplicate reads[53]. For dataset 4 containing experiments in replicates, we merged replicate BAM files and called peaks using MACS2 (version 2.1.0.20150731-Python-2.7.11) to retain peak scores[54], then BEDtools (version 2.26.0) was utilized to intersect peaks from experiments using dataset 1, which contained the highest number of peaks, as a reference with the output being the number of overlaps at each peak[55]. Peaks merged from experiments were then converted to DNA sequence in FASTA format with BEDtools and annotated by ChIPseeker (version 1.22.1) package in RStudio (version 3.6.3 used throughout the study)[56]. The FASTA sequences were then located to GATA1-binding sites by FIMO (MEME Suite version 5.0.4) and intersected with a set of GATA1-binding sites within the blood group genes (as defined in the Erythrogene database) called by JASPAR[57–59]. Transcription factor genes *KLF1* and *GATA1* were filtered out in a later step since the focus of the study is blood group genes. GATA1 matrix profile MA0035.3 was used for FIMO and JASPAR. The GATA1-binding sites were then merged with the peak annotation file and filtered with a list of blood group gene list and ranked by the number of overlapping peaks. The predicted targets of interest were then compared with databases gnomAD (version 3.1.2) and Ensembl GRCh38 (last accessed August 1^st, 2022) for variant annotation and frequencies[30,60]. Human genome build GRCh38/hg38 was used throughout the study. Lists of the JASPAR GATA1-binding sites, and the GATA1 matrix profile are also provided in the GitHub repository (https://github.com/ILM-MLOlab/ChIP-seq-GATA1)[61].

### Preparation of nuclear extracts

Nuclear extracts were prepared from K562 cells as described previously[62]. In brief, K562 obtained from the department's archive of frozen cell lines were cultured with RPMI1640 medium (Gibco) with

10% fetal bovine serum (Gibco). Cells were washed with ice-cold phosphate-buffered saline (PBS) and resuspended in hypotonic buffer (10 mM HEPES-KOH pH7.9, 1.5 mM $MgCl_2$, 10 mM KCl, 0.5 mM dithiothreitol) with 1× cOmplete protease inhibitor (Roche) and incubated on ice for 15 min. 10% NP-40 was added and the mixture vortexed and centrifuged at 800 × $g$ for 10 min at 4 °C. The supernatant was transferred to a clean, pre-chilled microcentrifuge tube. Protein content of the nuclear extract was quantified with the Pierce BCA kit (Thermo Fisher Scientific) and stored as single-use aliquots at −80 °C.

### Electrophoretic mobility shift assay (EMSA)

EMSAs were performed as previously described[3]. Double-stranded DNA probes were synthesized commercially (Eurofins Genomics) with biotinylation at the 5' end of both strands. Probes were prepared in Tris-EDTA (TE) buffer (Invitrogen) to the appropriate concentration and incubated with K562 nuclear extracts and (for supershifts) rabbit polyclonal anti-human GATA1 (Active Motif #61535). The reactions were prepared and then resolved on a 5% polyacrylamide Tris-borate EDTA gel (Bio-Rad) at 100 V for 1 h 15 min. Probes were electroblotted onto a Zeta-Probe membrane (Bio-Rad) with Trans-Blot Turbo (Bio-Rad) at 25 V for 30 min. The probes were UV-crosslinked to the membrane for 10 min with the ChemiDoc Touch Image System (Bio-Rad). Membranes were processed with the LightShift EMSA optimization and control kit (Thermo Scientific) and imaged with the ChemiDoc Touch. Three independent experiments were performed for each selected target predicted by the ChIP-seq analysis. Probe sequences are listed in Supplementary Data 3.

### Plasmid construction

The *CR1* promoter region (151 bp) was adapted from Kim et al., defined as bases −74 to +77 from the potential TSS of *CR1*[63]. This promoter sequence was amplified and cloned into the pGL3-basic vector at the *Bgl*II and *Hind*III restriction sites. Enhancer regions for motif 1 (437 bp), motif 2 (515 bp) or a combination of both motifs (952 bp) were cloned from healthy donors. Mutant for motif 1 sequence was made by overlap extension PCR using two primers containing the mutation as designed by Quikchange (Agilent)[64,65]. The enhancer regions were cloned into the vector with *Kpn*I to obtain both forward and reverse orientations. Constructs were verified by Sanger sequencing and plasmids were purified with the Plasmid MidiPrep Kit (Qiagen). Primers for this study are listed in Supplementary Data 3 and synthesized commercially (Eurofins Genomics).

### Luciferase reporter assay

Luciferase reporter assays were performed as previously published and a brief description follows[3]. For each experiment, $2 × 10^6$ HEL cells obtained from the department's archive of frozen cell lines were electroporated with 10 μg of either the pGL3-basic vector or the construct of interest, and 1 μg of pRL-null vector carrying the *Renilla* luciferase gene was co-electroporated as an internal control. The electroporation was carried out at 150 V, infinite resistance, over 10 ms in 1 square pulse in 0.2 cm gap cuvettes. HEL cells electroporated with no plasmids were used as negative controls. The firefly and *Renilla* luciferase activities were assessed with the Dual-Glo Luciferase Assay System (Promega) and measured with a GloMax Discover microplate reader (Promega) according to the manufacturer's protocol. All samples were run in technical triplicates and experiments were repeated thrice. The ratio between firefly and *Renilla* luciferase luminescence values were calculated using GraphPad Prism version 9.3.1, and the results were compared using two-sided paired T test on the means of three independent experiments.

### Genotyping of rs11117991, rs2274567, *Hind*III restriction site and haplotype frequencies of associated SNVs

We tested 100 anonymized EDTA blood samples for the Swedish cohort and 396 samples for the Thai cohort[66]. DNA was extracted from whole blood in EDTA tubes using automated DNA extraction modules (EZ1 Advanced and QIASymphony, QIAGEN) according to the user manuals. For *CR1*, we genotyped the SNV predicted in this study for the Helgeson phenotype, rs11117991, and the SNV previously reported to predict the Helgeson phenotype, rs2274567, by commercial predesigned TaqMan genotyping assays (Applied Biosystems). The *Hind*III restriction site was characterized with PCR-RFLP as previously described[67]. In short, samples were digested at 37 °C for 15 min with *Hind*III (Fermentas) followed by inactivation of *Hind*III at 65 °C for 20 min and resolved on 1% agarose gel electrophoresis. Confirmation of genotyping results of the TaqMan assay or the *Hind*III site on selected samples was carried out by a Sanger sequencing service (Eurofins Genomics). Haplotype frequencies of rs11117991 and rs2274567 from the two cohorts were calculated according to the observed genotypes. Samples heterozygous for both SNVs were assumed as the common haplotypes. Haplotype frequencies of rs11117991, rs2274567, and SNVs for the Knops antigen from 1000 Genomes project were calculated with the LDhap tool using LDlink version 5.3[31]. SNVs for the Knops antigen were included according to the International Society of Blood Transfusion (ISBT) allele table for the Knops blood group system (KN blood group alleles v4.0 31-MAR-2022)[68]. Measures of linkage disequilibrium (LD), $r2$ and $D'$, between the SNVs from the 1000 Genomes project were retrieved from LDpair tool using LDlink. Primers used in this study are listed in Supplementary Data 3 and synthesized commercially (Eurofins Genomics).

### RBC membrane preparation

Preparation for RBC membrane for western blot analysis was performed as described elsewhere[69]. Blood was washed in cold PBS at pH7.4 and lysed in 10 volumes of cold lysis buffer (4.65 mM $NaH_2PO_4$, 7.2 mM $Na_2HPO_4$) with 0.1× cOmplete protease inhibitor (Roche). Samples were then vortexed immediately and incubated on ice for 10 min before centrifugation at 4 °C, 16,000 × $g$ for 10 min. The supernatant was removed and the lysis process was repeated until pellets became pale, indicating that the majority of hemoglobin was removed. Samples were stored −80 °C.

### Immunoblotting

RBC membranes were boiled in Laemmli buffer (Bio-Rad) for 2 min at 99 °C and resolved in 7.5% Mini-Protean TGX Stain-free gels (Bio-Rad) with the Dual Color protein standard (Bio-Rad) under 200 V for 1 h. The gel was then activated by ChemiDoc Touch Image System (Bio-Rad) and an image of total protein content was recorded. Next, proteins were transferred to a polyvinylidene fluoride (PVDF) membrane with the Trans-Blot Turbo Transfer System and Mini PVDF Transfer Pack (Bio-Rad); the settings were modified to constant 1.3 A, 25 V for 15 min to enhance transferring of larger proteins. The membrane was then blocked with 5% w/v dry milk in TBS-T (20 mM Tris base, 137 nM NaCl, 0.1% Tween 20) buffer at room temperature for 1 h with constant shaking, followed by washing with TBS-T buffer. The primary antibody, mouse anti-human CD35 (clone E11, Bio-Rad, #MCA554GA), was added at 1:1000 dilution in 0.5% dry milk in TBS-T and gently agitated at 4 °C overnight. After washing, the secondary antibody, polyclonal goat anti-mouse IgG1 HRP (Bio-Rad, #1706516), was added at 1:3000 dilution in 0.5% dry milk with TBS-T and incubated for 1 h at room temperature with gentle agitation, and followed by washing. All washing was performed with TBS-T first with 15 min under gentle agitation, and then 5 min with gentle agitation twice more. Chemiluminescence detection was performed with the Clarity Western ECL substrate (Bio-Rad) following the manufacturer's protocol and images were taken with ChemiDoc Touch Image System. The intensity of the band of interest of each sample was normalized to its own total protein content with Image Lab (Bio-Rad, version 6.0.1)[70]. The normalized intensity of CD35 was then compared between the groups with one-way ANOVA and tested post-hoc with the Bonferroni method in RStudio.

## Flow cytometry

CR1 antigen expression was analyzed on RBCs from 100 donors in the Swedish cohort and 41 donors from the Thai cohort subset. In the latter, 11 samples were selected that carried haplotype 3 (Fig. 4), and 10 samples by random selection from each group of genotypes of homozygous major, heterozygous and homozygous minor for both rs11117991 and rs2274567. Briefly, 5 µl of 3% RBC suspension was added to 50 µl of PBS + 2% bovine serum albumin, followed by 5 µl of PE-conjugated anti-CD35 (clone 11, BD Pharmingen, #559872) in a round bottom 96-well plate. The plate was incubated for 30 min at room temperature with continuous shaking at 300 rpm in the dark. The cells were centrifuged for 1 to 3 min at $300 \times g$ and washed twice with PBS, then resuspended in PBS and analyzed on a FACSCanto (BD) flow cytometer with BD FACSDiva version 8.0.1 software or Amnis Cell-Stream (Luminex) with CellStream Analysis version 1.4.72 for the Swedish and Thai cohorts, respectively. The gating strategy can be found in Supplementary Fig. 2. The geometric mean fluorescence intensity (MFI) was compared between each genotype group with one-way ANOVA and post-hoc test with the Bonferroni method in RStudio. Multiple linear regression analysis for the Thai cohort was performed in RStudio, assigning the following variables for the genotypes: the homozygous major is designated as 0, heterozygous as 1, and homozygous minor as 2.

## *CR1* gene expression

RNA extraction was performed as follows. RBCs frozen in glycerol were thawed and passed through leukocyte syringe filters (PALL). One hundred microlitres of packed RBCs was added to 750 µl of TRIzol LS (Ambion, Life Technologies), then incubated for 5 min at room temperature or stored at 4 °C overnight. Next, 200 µl of chloroform (MP Biomedical, LLC) was added, inverted to mix, and incubated for 3 min at room temperature. The samples were then centrifuged for 15 min at $12,000 \times g$ at 4 °C, and the aqueous phase containing RNA was then transferred to a clean tube. An equal volume of 70 % ethanol was then added, mixed well, and transferred to a Mini Elute Spin column from the RNeasy Plus Mini kit (QIAGEN), and the RNA was purified following from step 4 and onwards in the manufacturer's protocol. In brief, the columns were spun down >15 sec, $>8000 \times g$, with the flow-through discarded and followed by washing with 750 µl of RW1 buffer and twice with 500 µl RPE buffer. Lastly, the columns were centrifuged at full speed for 1 min before placing into a new collection tube, after which RNA was eluted in nuclease-free water. The cDNA synthesis was done with SuperScript IV VILO (Invitrogen) without DNase treatment following the manufacturer's protocol. Gene expression of *CR1* was measured in a duplex reaction using TaqMan Gene expression assays (Thermo Fisher) for FAM-labeled *CR1* and VIC-labeled *GAPDH* probes, where *GAPDH* served as the internal control. All experiments were run in triplicate with Taqman FastAdvanced Master Mix (Thermo Fisher) on a QuantStudio 3 real-time PCR system (Applied Biosystems) and analyzed with QuantStudio Design & Analysis version 1.5.1. The mean of the triplicates was calculated and the ΔΔ CT was performed using *GAPDH* as the internal control and an inter-plate sample of *CR1* as the normalizing control. Multiple linear regression analysis for the Thai cohort was performed in RStudio with variables assigned as stated above.

## Reporting summary

Further information on research design is available in the Nature Portfolio Reporting Summary linked to this article.

## Data availability

The processed data of Jaspar motifs are available at github repository (https://github.com/ILM-MLOlab/ChIP-seq-GATA1/tree/main/JasparFunctions). The predicted targets generated in this study are provided in the Supplementary Data 1. The raw sequencing reads from ChIP-seq experiments analyzed in this study were obtained from publicly available databases (GSE36985; GSE32491; GSE31477; ENCSR000EXP), refer to Supplementary Data 2 for detailed information on these experimental datasets. The sequencing data generated in this study have been deposited in the European Nucleotide Archive (ENA) under the project code PRJEB64594. Source data are provided with this paper.

## Code availability

Code used in this study is provided in the GitHub repository (https://github.com/ILM-MLOlab/ChIP-seq-GATA1; https://zenodo.org/badge/latestdoi/501283332)[61].

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

## Acknowledgements

We thank Dr. Joann Moulds for fruitful discussions on CR1 biology, biochemistry and genetics, and for supplying the index Helgeson DNA sample, Professor Mikael Sigvardsson for the pGL3-basic and *Renilla* plasmids, Professor John Semple and Dr. Geneviève Marcoux for their assistance with the Luminex Amnis flow cytometer, Dr. Philaiphon Jongruamklang for providing the Thai blood donor samples, Dr. Magnus Jöud for helpful discussions on bioinformatics and the Department of Clinical Immunology and Transfusion Medicine in Region Skåne for the provision of samples from Swedish blood donors and DNA extraction for both cohorts. This study was supported by the Knut and Alice Wallenberg Foundation (2014.0312 and 2020.0234 to M.L.O.), the Swedish Research Council (2019-01683 to J.R.S. and M.L.O.) and governmental ALF grants to the university healthcare in Region Skåne, Sweden (ALFSKANE-446521 to M.L.O.).

## Author contributions

P.C.W., M.M., and M.L.O. conceived and designed the bioinformatic part of the study, whilst P.C.W., Y.Q.L., J.R.S., and M.L.O. designed the in vitro experiments of the study. P.C.W. and M.M. performed the bioinformatic analysis, P.C.W. and Y.Q.L. performed experiments. All authors interpreted bioinformatic and in vitro experimental data. Y.Q.L., J.R.S., and M.L.O. supervised the study. P.C.W. wrote the manuscript and all authors read and revised the manuscript.

## Funding

## Competing interests

The authors declare no competing interests.
