## [Peer Review File · Nature Communications]

Elucidation of the low-expressing erythroid CR1 phenotype by bioinformatic mining of the GATA1-driven blood-group regulomeREVIEWER COMMENTS

Reviewer #1 (Remarks to the Author):

The authors report on the identification of GATA1 binding sites in blood group genes by in silico and experimental data. Using CR1 (low-expressing CR1 for the Helgeson phenotype; Knops blood group system) as a reasonable model it was clearly shown that a GATA1 site in intron 4 is associated with the mRNA and protein expression level of CR1.

Comments:

- 1) The measurement of CR1 expression on RBCs showed a broad range of values (Figure 5). Especially samples with the rs11117991-TT genotype and low expression values within the lower 1.5 times of the interquartile range could have other genetic markers lowering CR1 expression. Has the CR1 region including motifs 1 and 2 been sequenced in these samples?
- 2) Alloantibodies to Kn antigens have rather no clinical significance but are frequently observed in transfused patients. It would be interesting to see whether the rs11117991 genotype is associated with Kn antibody formation in patients.
- 3) Fig. 6: the different order of genotypes in the dot blots is confusing. Please use the same order in both dot blots, i. e. in a: CC – TC – TT and in b: TC – TT).
- 4) Please add further information about the ChIP-seq data (ethnic background of the 4 data sets).

Reviewer #2 (Remarks to the Author):

The paper by Wu et al uses a functional genomics approach to identify the genotypes associated with the Helgeson rare blood group antigen which is characterized by very low level of expression of CR1. This rare blood phenotype may be erroneously interpreted as CR1 null, resulting in the development of immunoreactions upon repeated transfusion. The paper identified that the phenotype is associated with two SNPs, which alter the GATA1 binding sites in the regulatory regions of the CR1 gene. Mechanistically, the two SNPs were identified to reduce the expression of reporter genes driven by the CR1 regulatory sequences in relevant cell lines. The data are robust, properly discussed and of clinical interest for the field of Transfusion Medicine. It is however recommended that the discussion includes previous examples of blood group definition by GWS (PLoS One. 2022; 17(6): e0269481) and of additional blood group antigens already reported to be associated with loss of erythroid-specific

transcription factor functions (Ernstman et al Sci Rep. 2021; 11: 18557 and Singleton et al Blood 112(5), 2081–2088 (2008).

Reviewer #3 (Remarks to the Author):

I checked the source code repo, and found the information is to be incomplete. The authors should upload more scripts to ensure that readers can reproduce the bioinformatics analysis.

Although the script is relatively small, I still found some mistakes. For example, the ChIPseq analysis script, <https://github.com/ILM-MLOlab/ChIP-seq-GATA1/blob/main/analysis2022.R>, uses the ChIPseeker package (<https://currentprotocols.onlinelibrary.wiley.com/doi/epdf/10.1002/cpz1.585>). But in the article and Figure 1, it is incorrectly stated as the ChIPpeakAnno package.

Please double check.

Reviewer #4 (Remarks to the Author):

The authors present an *in silico* process to identify GATA-1 motifs that drive expression of blood group genes. The process is sufficiently detailed and supported with follow-up *in vitro* work on CR1 provided support for the model to examine some 190 or more motifs for the effect on other blood group genes. The discussion is well written, with the limitations of the study being delineated as elements of the CR1 analysis are summarized.

Top of page 11. An additional four GATA motifs did not fit the reported Helgeson prevalence (which is the focus of the analysis). The role of these additional motifs was not investigated at this time (rationale given), and the authors outline the need to have a complete understanding of the regulation of CR1. The selective pressure of malaria is key to the SNV of CR1 (page 13 addresses malaria in this context). It would be worth stating that the influence of malaria makes additional analysis worthwhile on page 11 or merge the information on page 13.

Minor comments and edits

The abstract should contain the location of the GATA-1 motifs studied (intron 4).

Page 2, line 26. "... enhancer motifs 'that' bind GATA-1..."

Page 10, line 216. Cite the two basepair regions containing the GATA-1 motifs.

Page 10, line 224. "... of CR1 stood out. How did the two motifs stand out?"

REVIEWER COMMENTS ON NCOMMS-22-47756 (Wu et al.)

Reviewer #1 (Remarks to the Author):

The authors report on the identification of GATA1 binding sites in blood group genes by in silico and experimental data. Using CR1 (low-expressing CR1 for the Helgeson phenotype; Knops blood group system) as a reasonable model it was clearly shown that a GATA1 site in intron 4 is associated with the mRNA and protein expression level of CR1.

Reviewer 1's comment	Response	Edits on page/line
The authors report on the identification of GATA1 binding sites in blood group genes by in silico and experimental data...	Thank you.	-
1) The measurement of CR1 expression on RBCs showed a broad range of values (Figure 5). Especially samples with the rs11117991-TT genotype and low expression values within the lower 1.5 times of the interquartile range could have other genetic markers lowering CR1 expression. Has the CR1 region including motifs 1 and 2 been sequenced in these samples?	Thank you for this question: We have sequenced a 952-bp fragment of the CR1 region surrounding and including motifs 1 and 2 of the samples mentioned by the reviewer (rs11117991:T/T genotype with the lowest expression values in Fig. 5). The sequenced fragment was amplified with the primers mentioned in Suppl. Table 5 for luciferase assay plasmid construction. In total, seven samples (5 Swedish and 2 Thai) were tested as requested:  1. Three samples had no deviations from the CR1 reference sequence (NG_007481), although all samples carried rs10779311:C. However, its frequency in gnomAD is 0.9855. 2. One Swedish sample was heterozygous for rs61822967:G>A, which has a frequency of 0.2526 according to gnomAD. 3. Three other samples (two Swedish sample and one Thai) were heterozygous for rs12043913:G>T at 0.2512. 	Page 9, lines 185-192 & Page 13, line 293

	4. One (Thai) of the latter samples was also heterozygous for rs147061134:A>C, a SNV with a frequency of 0.0125 according to the 1000 Genomes project and 0.0248-0.0316 in Asian populations (but not found in gnomAD). 5. None of the above SNVs disrupt or are found adjacent to the two GATA1 motifs 1 and 2, nor did they interfere with binding sites for other key erythroid transcription factors. In conclusion, we did not reveal any plausible explanation underlying the low expression in the rs11117991:T/T genotype group. In the manuscript, we have now reported the findings summarized above in the Results section and drawn attention to Fig. 5 in the Discussion where variation was already mentioned.	
2) Alloantibodies to Kn antigens have rather no clinical significance but are frequently observed in transfused patients. It would be interesting to see whether the rs11117991 genotype is associated with Kn antibody formation in patients.	To our knowledge, there is no literature supporting a correlation between Helgeson phenotype and alloantibody production against CR1 in patients. We also consulted Dr. Joann Moulds (mentioned in acknowledgements), one of the world experts on CR1 immunohematology, who agreed with this. The Helgeson phenotype as defined by GATA1 binding disrupted by rs11117991:C confers very low levels of CR1 on red cells, but GATA1 does not affect expression on other cells. Instead, alloantibodies in the Knops system are due to the complete absence of the specific alloantigens as encoded by single amino acid changes in CR1. In the absence of access to samples from a cohort of transfused patients	Page 15, line 348-351

	with antibodies against Knops antigens, we have now incorporated a sentence discussing the reviewer's point about the relationship between Helgeson and Knops antibody formation.	
3) Fig. 6: the different order of genotypes in the dot blots is confusing. Please use the same order in both dot blots, i. e. in a: CC – TC – TT and in b: TC – TT).	Thank you for this important comment. We have now revised the figure accordingly to make it easier for readers to follow. Please observe that the red color used is color-blind proof.	Figure 6 and legend
4) Please add further information about the ChIP-seq data (ethnic background of the 4 data sets).	Information has been added to Supplementary Table 4. However, no ethnicity information was disclosed in the original articles so cannot be provided, unfortunately.	Supplementary Table 4

Reviewer #2 (Remarks to the Author):

The paper by Wu et al uses a functional genomics approach to identify the genotypes associated with the Helgeson rare blood group antigen which is characterized by very low level of expression of CR1. This rare blood phenotype may be erroneously interpreted as CR1 null, resulting in the development of immunoreactions upon repeated transfusion. The paper identified that the phenotype is associated with two SNPs, which alter the GATA1 binding sites in the regulatory regions of the CR1 gene. Mechanistically, the two SNPs were identified to reduce the expression of reporter genes driven by the CR1 regulatory sequences in relevant cell lines. The data are robust, properly discussed and of clinical interest for the field of Transfusion Medicine.

Reviewer 2's comment	Response	Edits on page/line
The paper by Wu et al uses a functional genomics approach...	Thank you for the positive comment.	-
It is however recommended that the discussion includes previous examples of blood group definition by GWS (PLoS One. 2022; 17(6): e0269481) and of additional blood group antigens already reported to be associated with loss of erythroid-specific transcription factor functions (Eernstman et al Sci Rep. 2021; 11: 18557 and Singleton et al Blood 112(5), 2081–2088 (2008).	Thank you for this suggestion. We have now included two sentences to go with the recommended references in the Discussion section.	Page 10, line 215-218 and refs 36-39.

Reviewer #3 (Remarks to the Author):

Reviewer 3's comment	Response	Edits on page/line
I checked the source code repo, and found the information is to be incomplete. The authors should upload more scripts to ensure that readers can reproduce the bioinformatics analysis.	Thank you for this important comment. We now have updated our GitHub page, separating the codes originating from different platforms and also written a detailed description of the analysis flow in the readme file.	GitHub repository has been revised as requested
Although the script is relatively small, I still found some mistakes. For example, the ChIPseq analysis script, https://github.com/ILM-MLOlab/ChIP-seq-GATA1/blob/main/analysis2022.R, uses the ChIPseeker package (https://currentprotocols.onlinelibrary.wiley.com/doi/epdf/10.1002/cpz1.585). But in the article and Figure 1, it is incorrectly stated as the ChIPpeakAnno package.	Thank you very much for spotting this error. The ChIPpeakAnno package was originally used in the "annotatePeakInBatch" function, which was later removed and replaced by 'annotatePeak" from the ChIPseeker package in our analysis. We now have revised the figures (both in the manuscript and the GitHub repository), the text and the reference accordingly. We also carefully checked all scripts/packages used to ensure the information is correct.	Changes to Figure 1, and the figure in the Github repository. Text edit on page 18, line 405 and change of ref. 56.

Reviewer #4 (Remarks to the Author):

The authors present an *in silico* process to identify GATA-1 motifs that drive expression of blood group genes. The process is sufficiently detailed and supported with follow-up *in vitro* work on CR1 provided support for the model to examine some 190 or more motifs for the effect on other blood group genes. The discussion is well written, with the limitations of the study being delineated as elements of the CR1 analysis are summarized.

Reviewer 4's comment	Response	Edits on page/line
The authors present an in silico process ...	Thank you for the positive comment.	-
Top of page 11. An additional four GATA motifs did not fit the reported Helgeson prevalence (which is the focus of the analysis). The role of these additional motifs was not investigated at this time (rationale given), and the authors outline the need to have a complete understanding of the regulation of CR1. The selective pressure of malaria is key to the SNV of CR1 (page 13 addresses malaria in this context). It would be worth stating that the influence of malaria makes additional analysis worthwhile on page 11 or merge the information on page 13.	The frequencies of the minor allele disrupting the GATA1 motif at the four other candidate sites, which were not further investigated, are all very low (1 to 3 alleles out of a total of >150,000 alleles in all superpopulations in gnomAD). Furthermore, these frequencies are not higher in malaria-affected populations according to gnomAD. Thus, they do not fit the Helgeson phenotype frequency but also do not favor the hypothesis of an obvious role in P. falciparum malaria leading to evolutionary selection. Instead, other target regions worth investigating in the future in CR1 may play such roles. Also, the qualitative principle (i.e. changing CR1-related antigens like Sla) may play such a role as discussed on p. 13-14. We added a sentence on p. 11 to clarify to readers the above and kept the malaria discussion mainly on p. 13.	Page 11, lines 250-254
Minor comments and edits The abstract should contain the location of the GATA-1 motifs studied (intron 4).	The information about the location of the motifs in intron 4 is now incorporated in the abstract.	Abstract, page 2, line 26
Page 2, line 26. "... enhancer motifs 'that' bind GATA-1..."	We added a "that" to this sentence.	Abstract, page 2, line 25

Page 10, line 216. Cite the two basepair regions containing the GATA-1 motifs.	The reason why this information was not included here is that the exact rs numbers and locations of the two motifs are given in the Results section (p. 6) and in the Suppl. Table 1, respectively. We have now added the rs numbers for the variants disrupting the two motifs, also in the Discussion.	Page 11, line 237-238
Page 10, line 224. "... of CR1 stood out. How did the two motifs stand out?"	The underlying reason why motifs 1 and 2 were selected over the other four candidate motifs was given in the 2nd paragraph of the Results section but have now been reiterated and clarified also here.	Page11, line 235-236

In addition, we changed the order of the Data availability and Code availability sections at the end of the manuscript according to the guidelines (previously in reverse order). However, this change was not made in tracking mode.

REVIEWERS' COMMENTS

Reviewer #1 (Remarks to the Author):

All comments were considered in the revised manuscript.

Reviewer #3 (Remarks to the Author):

The authors have solved all my concerns.